# The Influence of External Parameters on the Ripeness of Pumpkins

**DOI:** 10.3390/s24010143

**Published:** 2023-12-27

**Authors:** Kubiat Emah, Linli Hu, Solomon Boamah, Sylvester Chukwuka, Richard John Tiika, Kai Zhang, Jianzhong Tie, Zhongqi Tang, Jihua Yu

**Affiliations:** 1College of Horticulture, Gansu Agricultural University, Lanzhou 730070, China; kubiatemah@outlook.com (K.E.); hull2022@163.com (L.H.); nlttzhangkai@126.com (K.Z.); kbcny1988@gmail.com (J.T.); imabasi2023@outlook.com (Z.T.); 2College of Plant Protection, Gansu Agricultural University, Lanzhou 730070, China; solomon4408boamah@gmail.com; 3Gansu Provincial Biocontrol Engineering Laboratory of Crop Diseases and Pests, Lanzhou 730070, China; 4College of Information Science and Technology, Gansu Agricultural University, Lanzhou 730070, China; syfreeman1992@gmail.com; 5College of Forestry, Gansu Agricultural University, Lanzhou 730070, China; tiikarichard@gmail.com

**Keywords:** greenhouse, environmental parameters, sensors, error-based, entropy-based approaches

## Abstract

Growing pumpkins in controlled environments, such as greenhouses, has become increasingly important due to the potential to optimise yield and quality. However, achieving optimal environmental conditions for pumpkin cultivation requires precise monitoring and control, which can be facilitated by modern sensor technologies. The objective of this study was to determine the optimal placement of sensors to determine the influence of external parameters on the maturity of pumpkins. The greenhouse used in the study consisted of a plastic film for growing pumpkins. Five different sensors labeled from A1 to A5 measured the air temperature, humidity, soil temperature, soil humidity, and illumination at five different locations. We used two methods, error-based sensor placement and entropy-based sensor placement, to evaluate optimisation. We selected A3 sensor locations where the monitored data were close to the reference value, i.e., the average data of all measurement locations and parameters. Using this method, we selected sensor positions to monitor the influence of external parameters on the maturity of pumpkins. These methods enable the determination of optimal sensor locations to represent the entire facility environment and detect areas with significant environmental disparities. Our study provides an accurate measurement of the internal environment of a greenhouse and properly selects the base installation locations of sensors in the pumpkin greenhouse.

## 1. Introduction

Originating in the Americas, the pumpkin, scientifically known as *Cucurbita moschata*, has become a significant crop in China, where it has been cultivated for approximately five centuries [1]. In China, it serves as both a staple food and a vegetable, demonstrating its adaptability to different environments and its ability to thrive throughout the country, resulting in substantial yields per unit. In addition to its fundamental role in alleviating famine during the summer and fall seasons as a cucurbitaceous vegetable, the pumpkin is also useful as livestock feed and as an ingredient in traditional Chinese herbal medicine [2]. Remarkably, China is the world’s largest consumer and producer of pumpkins, as reported by Abdel-Ghany and Al-Helal [2]. In the early 20th century, high-quality pumpkin breeds gained widespread recognition, with cultivation flourishing particularly in Sichuan Province and various regions of northern China [3]. This expansion of pumpkin cultivation had a profound impact on the rural economy and led to changes in farming systems, the alleviation of famine, and the enrichment of traditional Chinese medicine.

The growing of pumpkins in controlled environments, such as greenhouses, has gained recognition due to its potential to optimise yield and quality [4]. Growers can adjust a range of environmental factors, such as the soil humidity, CO_2_ levels, temperature, humidity, illumination intensity, and nutrient availability, to create the perfect growing environment for pumpkins in greenhouses [5]. Using sensor technologies in greenhouses to monitor the impact of both internal and external environmental parameters on vital crop growth and development is significant in modern smart agriculture practices. Precision agriculture plays a central role in modern agriculture, as it provides the ability to accurately determine the optimal seeding density, estimate the required amounts of fertilisers and other inputs, and make more accurate predictions of crop yields [6]. The foundation of precision agriculture relies on the continuous monitoring of several variables that include environmental factors (such as climate monitoring and control), economic considerations (including resource costs such as water, soil, and fertiliser), and human requirements (e.g., product market demand) [7]. Information technology, including satellites, sensors, global positioning system (GPS) technology, and aerial imagery, is critical to ensuring an accurate and continuous monitoring process. Numerous research papers have delved into crop management and the environmental factors that affect crop protection and highlight the central role of improved automation and efficiency [2]. A central contribution of these works is the development of hardware and software solutions to solve challenges related to climate control in greenhouses using wireless sensor networks. They also present methods for improving data security and conserving energy within the sensor nodes [8].

To assess the greenhouse environment, growers and researchers employ various sensors. However, due to economic constraints and complex management procedures, growers typically deploy a limited number of sensors. The choice of sensor locations is often based on the expertise of growers and greenhouse designers. Traditionally, a central position within facilities is considered a representative location for monitoring environmental factors like the air temperature and humidity [9]. However, uncertainty persists about whether measurements from the centre truly reflect the entire greenhouse environment. Consequently, it is essential to identify optimal sensor installation locations to effectively monitor the internal conditions of expansive greenhouses with a limited number of sensors. The ideal sensor placement has been determined based on criteria such as the z-index, outliers, standard deviation, and average. In the study by Chang et al. [10], the theory of information entropy was employed to identify the best combination of sensor locations for pressure measurement in a water pipe network. Utilising information entropy, sensor locations offering the highest amount of information were chosen as the optimal positions.

There are currently a large number of systems for monitoring crops. However, most of them are based on customised, individual solutions. A common characteristic among these solutions is the interdependence between the hardware and software components provided [11]. Consequently, it is extremely complex to introduce new hardware support and expand the functionality of the software tools [12]. Thus, replicating the outcomes of these systems in regions with different climatic and environmental conditions becomes a formidable challenge. Another inherent issue in crop monitoring systems is their limited adaptability to the ever-evolving requirements of plants, given that these necessities shift throughout various phenological stages [13]. Consequently, there is an urgent need for crop monitoring systems that are flexible and thus allow for the expansion of their functions to meet the dynamic and demanding requirements of the agricultural sector [14]. However, previous research has overlooked the consideration of time series of environmental data and the potential installation of multiple sensors within a greenhouse which is a limitation in current smart farming technology. Additionally, previous studies on agricultural facilities have selected optimal sensor locations based on subjective criteria such as minimal environmental change or maximum differences in environments, lacking quantitative measures. Despite efforts to choose locations that represent the entire facility environment, it is crucial to install sensors in areas that are significantly affected by unstable external conditions. This current study aims to determine the ideal sensor placement for precise control of the greenhouse’s overall environment and the detection of locations where there are significant fluctuations in air temperature, humidity, lighting, soil temperature, and air moisture due to external influences. The focus here is on data relating to environmental parameters, which comprise a critical factor in pumpkin growth, chosen for its ease of simultaneous measurement at multiple locations.

## 2. Materials and Methodology

### 2.1. Experimental Greenhouse Description

The greenhouse used for the experiment is located at 35.85° S and 104.08° N in Yu Zhong, Lanzhou, Gansu Province, China (Figure 1a). The structure is 57 m long, and the roof ridge extends from east to west. The ceiling is 5 m high, and the ridge is 4.8 m high. The greenhouse has two manually controlled vents at two different locations; on bright days, the vents are opened to prevent or reduce humidity in the greenhouse at night (Figure 1b). The greenhouse also has water pipes that are used to irrigate the seedbeds (Figure 1c). The heat-insulating cover is closed at 5 p.m. in winter and opened at 8 a.m. the next morning to let in sunlight. In summer, the vents and sunshade do not need to be closed, as this depends on the weather. The interior of the greenhouse is chosen as the numerical calculation area. The analytical approach and the computational domain are shown in Figure 2.

The greenhouse is 10 m wide, has a ridge height of 4.8 m, and measures 62 m outside and 57 m inside. This grid generation method uses the tetrahedral grid, which has a size of 0.2 m, a total grid size of more than 1.5 million, and an estimated skewness value of less than 0.5, occupying 98%. The greenhouse roof-covering materials are plastic film (outside) and aluminised plastic film (inner shading net). 

### 2.2. Plant Materials and Growth Conditions

The experimental site is situated in Lanzhou City, Gansu Province, China, at coordinates 35°87′ N and 104°23′ E. This region experiences a temperate semi-arid continental climate characterised by distinct seasons. It lies at an average altitude of 1790 m above sea level and has a mean annual temperature of 6.6 °C, with annual rainfall averaging between 300 and 400 mm. Evaporation is relatively high, with a mean annual evaporation rate of 1343 mm, and there is a frost-free period of around 150 days. The topography of the experimental field is gently sloped, and soil fertility is uniform throughout. The soil type in the experimental area is yellow soil, which is rich in calcium carbonate. The most important physico-chemical parameters of the topsoil (0–20 cm) included a phosphorus content of 131 mg kg^−1^, fast-acting potassium of 376 mg kg^−1^, organic matter of 5.31 g kg^−1^, electrical conductivity of 296 µS cm^−1^, and a pH level of 8.15. The test material used was the ‘Dong Feng No. 5′ variety of Zucchini. The conventional fertilisers used in the study consisted of urea (N ≥ 46%), calcium superphosphate (P_2_O_5_ ≥ 16%), and potassium sulfate (K_2_O ≥ 52%). The experimental design followed a completely randomised block design with three replicates for each treatment. The treatments included: T1: No fertiliser and T2: fertiliser at a ratio of 6.5 parts soil mixture (SM) to 3.5 parts TV (local feedstock composting formula). Each treatment consisted of three replicates, with each experimental plot covering an area of 43.2 square meters. The planting density was 18,890 plants per hectare, and the ridges were 80 cm wide, with a 60 cm-wide ridge surface. Furrows were 40 cm wide, the distance between the plants spacing was 100 cm, and the row spacing was 20 cm. Before planting, compost was applied as the base fertiliser at a rate of 6000 kg ha^−1^, following local organic fertiliser recommendations. The base fertiliser was spread evenly and tilled to a depth of 25–30 cm to prepare the ridges. Apart from fertilisation, all field management practices were the same for all treatments. For yield assessment, five plants from each replicate were harvested, and the total yield for each selected plant was recorded from the beginning of fruiting.

### 2.3. Environmental Parameters and Sensor Information and Placement

The environmental parameters that significantly influence pumpkin growth, such as the ambient temperature, humidity, illumination intensity, and soil moisture, were measured. The indoor data were automatically collected using “Small Horn” sensors (Research Center for Information Technology in Agriculture, Beijing, China), model NH002, with an in-built button battery and a communication mode consisting of a 4G cellular network, suitable for real-time monitoring of greenhouse environmental parameters. The air temperature indicator offers a temperature resolution of 0.1 °C, an accuracy of ±0.5 °C, and a measuring range from −10 to 60 °C. With a measuring accuracy of ±5% RH and a precision of 0.1% RH, the air humidity measurement spans from 0% to 100% RH. Soil temperature measurement ranges from −40 to 80 °C, with a measurement accuracy ±0.5 °C and a measurement resolution of 0.1 °C. Illumination measurement ranges from 0 to 200,000 Lux, with a measurement accuracy of +7% and measurement resolution of 1 Lux. The outdoor meteorological parameters included temperature, humidity, solar irradiance, ground pressure, and wind speed, which were automatically recorded by a weather station installed outside the greenhouse. 

The sensors were strategically placed at different locations within the greenhouse, as shown in Figure 2, to record the variations in internal environmental parameters. Five measurement points (A1–A5) were selected in order to find the best location for sensors that would allow accurate monitoring and management of the internal environment. The data collection took place in summer (July) and early winter (November) 2022. 

### 2.4. Evaluation Method for Optimal Sensor Placement

#### Entropy-Based Method

It is helpful to find the best sensor location to identify the places in greenhouses with a distinct microclimate, in addition to the sensor location for monitoring the entire greenhouse. Claude Shannon established information entropy as a field of research in 1948 [2]. If one probability of occurrence is 1, and another probability of occurrence is 0, then the information entropy has a minimum value of 0. If all probabilities of occurrence are equal to 1/q, then log_2_ Pi has the highest value (E).
(1)$$H(X)=\sum_{i=1}^{m}\begin{array}{c} \mathrm{Pi}\;\left(E\right){log}_{2}\mathrm{Pi}\left(E\right)\\ \;\;\left(0\leq H\left(X\right)\leq {log}_{2}P\left(E\right)\right) \end{array}$$
(2)$$H\left(X,Y\right)=H\left(X\right)+H\left(\frac{Y}{X}\right)$$
where H(X) is the data entropy determined by a single sensor, H (X, Y) is the total entropy of the data determined by two sensors, and H (Y/X) is the conditional data entropy data determined by two sensors. Pi(E) is the probability mass function of a variable, and m indicates the data range of parameters measured at a particular sensor. For every set of sensor locations, the total information entropy was determined using the following formula:(3)$$\sum_{i=1}^{n}T(X_{i,}X_{j,}\ldots X_{p})=H(X_{R})+\cdots +H(X_{P})+\sum_{i=1}^{n-p}\sum_{j=k}^{p}\begin{array}{c} H\left(X_{i},X_{j}\right)\\ \left(i\neq j\right) \end{array}$$
where $\sum_{i=1}^{n}T(X_{i,}X_{j,}\ldots X_{p})$ is the total entropy of the sensor combination, and n represents all sensors in the entire greenhouse; p is the number of selected sensors. $H(X_{R})+\cdots +H(X_{P})$ is the sum of the selected entropy values of the sensors and $\sum_{i=1}^{n-p}\sum_{j=k}^{p}\begin{array}{c} H\left(X_{i},X_{j}\right)\\ \left(i\neq j\right) \end{array}$ is the number of data sent from unselected sensors to selected sensors.

### 2.5. Determination of Optimal Sensor Positioning

The optimal sensor location to record the entire greenhouse environment was selected using the error-based approach. Statistical metrics such as root-mean-square error (RMSE) and mean absolute percentage error (MAPE) were also calculated to verify the accuracy of the collected data at the locations determined using the error-based method. The RMSE serves as a measure of the dissimilarity between the combined trend and the reference trend. However, there is no defined quantitative criterion for the evaluation of the RMSE. In contrast, the MAPE serves as a measure of predictive accuracy, which is expressed as a percentage of error. Therefore, the MAPE was used to assess the accuracy of the combined trends compared to the reference trend. The evaluation of the deviations between the reference trend and the combined trends, depending on the number of installed sensors, was performed using these two statistical indices. Equations derived from previous research studies were used to calculate RMSE and MAPE [15]. The RMSE, MSE, and MAPE statistical indices were calculated using an Excel spreadsheet to confirm the accuracy of the measured data at the location selected using the error-based equations
(4)$$MSE=\frac{\sum_{t=1}^{n}{\left(A_{t}-F_{t}\right)}^{2}}{n}$$
(5)$$RMSE=\sqrt{\frac{\sum_{t=1}^{n}{\left(A_{t}-F_{t}\right)}^{2}}{n}}$$
(6)$$MAPE=\frac{\sum_{t=1}^{n}\left(A_{t}-F_{t}\right)}{n}\times 100$$
where n is the total number of data points, A_t_ is the current value of the reference trend, and F_t_ is the value of the combo trend at a specific point in time. 

### 2.6. Analysis of Environmental Data

Descriptive statistics were used to investigate the main characteristics of the environmental data collected in summer and at the beginning of the winter season according to the placement of the sensors. Boxplots were used to examine and display the distribution and standard deviation of readings at each location from the statistical analysis, as well as missing data. Boxplots provide an easy way to visualise basic statistical indices such as the mean, first quartile, median, third quartile, and outliers. SPSS software (V16.0; Inc., Chicago, IL, USA) was used to perform one-way ANOVA and Duncan’s multiple range tests. To examine differences between data collected at different sensor locations, significance was set at *p* < 0.05. The boxplots presented in the results show significant differences [16]. 

## 3. Results and Discussions

### 3.1. An Analysis Using Descriptive Statistics for Data on Indoor Environments Gathered over Time

The microclimate in greenhouses, which includes air temperature, humidity, soil temperature, soil humidity, and illumination, influences how quickly and how well plants develop. In this study, data were recorded from sensor locations A1 to A5 in the greenhouse as a function of the time in summer and early winter. These measurements included air temperature, air humidity, soil temperature, soil humidity, and illumination. The measured internal environmental factors in the greenhouse showed irregular up-and-down patterns at all sensor locations. In the summer of 2022, the average values for air temperature were 21.58 °C, 20.35 °C, 21.10 °C, 21.10 °C, and 21.30 °C, respectively, at the sensor locations. However, the average values for air temperature at the beginning of winter (November) were 21.52 °C, 20.32 °C, 20.63 °C, 20.63 °C, and 20.92 °C, respectively. The internal temperature variation exhibited similar fluctuation in increasing and decreasing over the days (Figure 2a), respectively. 

The data for air humidity were plotted against the time for summer and early winter in the greenhouse. The internal environmental condition mentioned above showed unpredictable upward and downward trends across days of monitoring. The average air humidity values in summer were 72.86% 75.92%, 62.45%, 62.45%, and 72.33%. At the beginning of winter, however, average values of 87.10%, 90.79%, 75.19%, 75.19%, and 88.40% were measured. 

Figure 3C shows a time-dependent representation of the soil temperature data in the greenhouse during the two seasons. The internal soil temperature showed irregular up-and-down patterns over the days during the study period, with the soil temperature starting to increase at the end of July and decreasing at the end of November. In summer, the corresponding average values for the soil temperature of the sensors were 20.85 °C, 20.28 °C, 19.80 °C, 19.80 °C, and 20.18 °C, respectively. Interestingly, the average values for the soil temperature at the beginning of winter were 21.72 °C, 20.89 °C, 20.42 °C, 20.42 °C, and 21.17 °C, respectively.

Soil humidity is one of the important factors influencing plant growth. Soil humidity data were recorded as a function of the time for both seasons. All sensors in the greenhouse showed an irregular upward and downward trend in soil humidity throughout the study period (Figure 3D). In comparison, the average soil humidity of the sensors was 17.64%, 20.13%, 16.90%, 18.36%, and 16.58%, while in early winter, average values of 20.13%, 18.44%, 17.86%, 17.86%, and 15.39% were recorded. 

The interior illuminance of the greenhouse also showed irregularly increasing and decreasing patterns over the course of the days in both seasons. In mid-summer, the average illuminance values recorded by all sensors were 6621.90 Lux, 6166.10 Lux, 7423.24 Lux, 7423.20 Lux, and 13,605.80 Lux, respectively. Also, the average values for early winter were 5825.96 Lux, 4221.70 Lux, 3799.98 Lux, 3799.00 Lux, and 10,201.64 Lux, respectively (Figure 3E).

### 3.2. Descriptive Statistical Evaluation of Sensor A1–A5 Data from Indoor Environments

Table 1 shows that although the temperature in the greenhouse averaged 19.67 °C in early winter, it occasionally exceeded 27.03 °C. According to Liu et al. [17], the highest temperature in the double-film greenhouse was 28.18 °C, the maximum temperature difference was 12.78 °C, and the conventional single-foil greenhouse had a maximum temperature of 15.4 °C. In Yu Zhong County, Lanzhou, Gansu Province, the mean and minimum air temperatures in the double-foil solar greenhouse are 6.92 °C and 5.51 °C higher than those in the single-foil solar greenhouse, respectively. The ideal range of air temperatures for pumpkin growth was 20–30 °C in both seasons. There are constant climate disturbances in the greenhouse, both inside and outside. Air flow, solar radiation, and other variables can significantly alter the results of the sensors in some locations (Figure 4). Fruit set has been found to occur at 22–26 °C, leaf and rod development at 22–25 °C, and pumpkin growth and fruit set occurs at 22–26 °C [2]. Warm-season greenhouse plants are usually acclimatised to air temperatures betweeen 17 °C and 27 °C, ranging from 10 °C to higher- and lower-limit temperatures of 35 °C [18], which is consistent with our results.

The temperature in the root zone also influences plant growth, especially during the development of new shoots and flowering. However, there may be an interaction with air temperature, solar radiation, and day length. The root zone temperature of 23 °C in summer had the least effect on drought stress in pumpkins, while treatments between 22 and 33 °C indicated that a root zone temperature of 27 °C would be best for pumpkin evaluation with the statistical methods used for comparative analysis [19]. Nevertheless, all five sensors showed that the soil temperature was higher in early winter than in the summer. Similarly, research on the average soil temperature in the double-foil greenhouse and standard single-foil greenhouse also showed that it fluctuated over time [17]. In the double-foil solar greenhouse, the soil temperatures at depth are lower throughout the day than in the conventional single-foil greenhouse, and the variance in soil surface temperature is significantly greater in the double-foil greenhouse than in the conventional greenhouse. The discrepancy in our results could be due to the type of greenhouse. As the soil thickness increases, the temperature also increases, which leads to a reduction in the temperature gradient between the different soil levels in greenhouses. Other research found that the temperature of the soil in a normal greenhouse was about 10.56 °C, with a difference of 6.95 °C between different areas. 

The greenhouse cover collects water droplets, while a portion of the moisture escapes through vents, and the plants and soil emit water vapor, contributing to an increase in the humidity level encapsulated within the greenhouse. However, in early winter, the indoor humidity values for A2 and A4 were significantly higher than 90%. The indoor temperature and the ventilation system of the greenhouse could be responsible for the differences in humidity in different seasons. According to [20], a relative humidity between 60 and 90% is suitable for most plants in the greenhouse. Pumpkins should have humidity levels between 50 and 70% while they are growing. Despite these previous studies, sensor A3 in this study recorded the ideal relative humidity as 62.46% in summer and 75.19% in early winter (Table 1). According to a study, pollination is greatly improved when the relative humidity is around 60%, which is consistent with our findings that using an A3 sensor provides optimal humidity readings [21]. It is important to emphasise that plants require more moisture when exposed to higher temperatures [19].

The soil humidity at a 10 cm depth in the double-foil solar greenhouse averaged 77.71% per day, while the soil humidity at the same depth in the single-foil greenhouse averaged 88.11% per day, which is 10.4% higher than in the double-foil greenhouse [17]. At a depth of 20 cm, the average daily soil humidity content is 62.53%, while in a typical single-layer greenhouse, it is 74.83%. This represents a moisture difference of 12.53%. The average soil humidity values of the soil layers in the two greenhouses with equivalent depths of 30 cm and 60 cm varied by 13.30%, 12.84%, and 12.79%, respectively. Similar to the previous study, the average differences in soil humidity during summer for sensors A1-A5 were 17.71%, 18.89%, 18.36%, 19.79%, and 16.58%, respectively (Table 1). In addition, sensors A1-A5 recorded soil humidity values of 20, 13%, 18, 44%, 17, 86%, and 15, 39% during the early winter season. 

One of the most important tasks in greenhouse design today is to reduce energy consumption for assimilation illumination. Lately, many ways to illuminate greenhouses have been tested. However, it is hard to tell how well they work in real life [22]. A hotly contested issue in greenhouse horticulture is the geographic placement of assimilation illumination, especially in countries with insufficient access to natural sun illumination [23,24,25]. Since most plants are illumination hungry, the amount, quality, length, and distribution of illumination have a significant impact on how well a pumpkin crop is cultivated in a greenhouse. Compared to other sensors, sensors A3 and sensor A5 each reported more illumination in the summer and similar illumination in the early winter. It is recommended that an additional illumination source be provided to supplement natural illumination. Considering the natural illumination reaching the plants, additional illumination could be achieved in the greenhouse. The geographical location, the season, the time of day, the illumination transmittance of the greenhouse glass, the shaded areas inside the building, and other factors can all influence these findings. 

### 3.3. Optimal Sensor Placements (A1–A5)

The ANOVA test was used to determine whether there were significant differences in the measured data between the sensor locations. The boxplots in Figure 5 show the distribution of data measured at each site for air temperature, soil temperature, air humidity, soil humidity, and illumination. According to the results of the ANOVA test, a statistically significant (*p* < 0.05) difference was found in air temperature, soil temperature, air humidity, soil wetness, and illumination measured at different locations in the greenhouse. The sensor locations can be categorised into three groups for summer and two groups for early winter, according to Duncan’s multiple-range test. The statistical analysis revealed significant (*p* < 0.05) variations in air temperature, soil temperature, air humidity, soil humidity, and illumination at several locations in summer and early winter. The best sensor location can be chosen, as seasonal variations in the internal environment of the greenhouse can affect the data needed to control the internal environment. 

To determine which sensors were the best, we compared the air temperature data from different sensors with the average air temperature data from all the sensors (A1–A5). We used three methods (MSE, RMSE, and MAPE) to test how well the sensor data matched the expected trend, as shown in Table 2. In general, MSE, RMSE, and MAPE were lower in summer than in early winter, meaning that the data collected at each sensor agreed better with the average in summer than in early winter. This is because there was more natural ventilation in summer, resulting in a more even distribution of air temperature in the greenhouse [2]. In the early winter, however, there were significant differences between the air temperature data recorded at each sensor and the reference trend because heat pumps were used for heating at night and natural ventilation during the day. Since the RMSE, MSE, and MAPE values were lowest at A3 and A5 (0.29 °C, 0.54 °C, and 2.00%, respectively), the data measured in summer at these two sites were most consistent with the reference trend. Like the summer data, the early winter data at A3 and A5 were closest to the reference trend, with their RMSE, MSE, and MAPE being the lowest (0.70 °C, 0.88 °C, and 3.41%), respectively. However, as natural ventilation was used for ventilation during the day, and heat pumps were utilised for heating at night, there were some significant discrepancies between the air temperature data recorded by each sensor and the reference trend in winter. As the MSE, RMSE, and MAPE values were lowest at A3 and A5 (0.29 °C, 0.54 °C, and 2.00%, respectively), the data measured in summer at these two sites matched the reference trend. Considering that the entire internal environment of the greenhouse was accurately represented in the summer (error = 2.12%) and early winter (error = 2.44%) with respect to the reference trend, it can be concluded that A3 measures soil temperature data much more accurately than A5 and the other sensors (Table 2).

The MSE, RMSE, and MAPE of A1, A2, and A4 were greater than the reference trend for air and soil humidity in both seasons (Table 2). The MSE, RMSE, and MAPE values for A3 and A5 for air moisture were lowest in summer and closest to the reference trend in winter (0.71%, 0.81%, and 1.05%, respectively). Similarly, MSE, RMSE, and MAPE of soil humidity for summer and early winter were well represented for A3 and A5 and were closer to the reference trend (0.31%, 0.55%, and 2.64%, respectively). However, the A3 sensor for indoor humidity throughout the greenhouse reflected the reference trend well (1.05% and 1.08% MAPE in summer and early winter, respectively) and should be used preferentially over A5. Based on the same explanation for soil humidity, sensor A3 was well represented for soil humidity inside the entire greenhouse concerning the reference tendency in summer and early winter (2.64% and 2.60% MAPE, respectively). 

In addition, compared to the reference trend, the illumination MSE, MAPE and RMSE of A1, A2, and A4 were greater in both seasons (Table 2). Similarly, the MSE, RMSE, and MAPE of summer illumination for A3 and A5 were well represented and closer to the reference trend (3235.93 Lux, 56.89 Lux, and 0.91% Lux, respectively) and early winter illumination (34,260.75 Lux, 158.62 Lux, and 3.51% Lux, respectively). Our results suggest that sensor A3, located in the centre of the greenhouse, is the best option for measuring indoor illumination after optimising the sensors with the MSE, RMSE, and MAPE approaches. Our results agree with those of [26], in which it was found that it is sufficient to place the sensor in the centre of the greenhouse and that it is crucial to select the right sensor positions according to greenhouse design and management strategies. Even if sensors are used at different locations, it is necessary to analyze the placement of the sensors objectively.

### 3.4. Evaluation of Optimal Sensor Placement (A1–A5)

The air temperature, soil temperature, air humidity, soil humidity, and illumination data obtained at five sensor locations were subjected to the error-based methodology and the entropy-based methodology to select the best sensor placement (Table 3 and Table 4). When only one sensor was used, the error-based approach showed that A3 was the ideal location for detecting air temperature, soil temperature, air humidity, soil humidity, and illumination in the greenhouse. The best locations for recording air temperature, soil temperature, air humidity, soil wetness, and illumination in the greenhouse while using two sensors were A3 and A5. The summer and early winter data show a clear trend for the entropy-based method compared to the error-based method. This is due to the fact that the region favours those data that have high data volatility and few duplicate data. Due to their higher information entropy than other locations, A3 and A5 were selected. Nonetheless, the final three were chosen: A1, A2, and A4. Due to the principal wind directions, which were influenced by the external wind conditions, A3 was chosen for summer instead of A5 (Table 3).

Table 4 shows the results for the optimal placement of the sensors in early winter based on the number of sensors. When one sensor is installed in the greenhouse, the error-based method showed that A3 is the best location for monitoring air temperature, air humidity, soil temperature, soil humidity, and illumination. The best locations for recording air temperature in the greenhouse with two sensors were A1 and A4. Due to low information entropy at A2, A3, and A5, these locations were chosen last in the entropy-based method. Consequently, the edges of the greenhouse were chosen as the ideal locations for the sensors. The distribution of environmental factors on both sides of the greenhouse showed significant differences due to the kind of covering material, ventilation techniques, and natural solar radiation. Our study has shown that the data obtained at the sensor locations for air temperature, soil temperature, air humidity, soil humidity, and illumination can be selected, evaluated, and used to generate values that are close to the reference trend. To choose sensor locations that accurately reflect the overall greenhouse environment, an error-based technique was created. On the other hand, error-based and entropy-based approaches can complement each other when used together. These results support the findings of [27,28], in which it was reported that the coefficient of determination, MSE, MAE, and RMSE assessment models, and simulated results indicated that the multiple linear regression projection (MLR) models showed high correlation with the measured data with lower RMSE, MSE, and MAE.

### 3.5. The Influence of Outdoor Environmental Conditions on the Microclimate in Greenhouses

To determine the influence of the outside environmental climate on the indoor microclimate, the data from the A3 sensor were used for correlation analysis. Because the greenhouse was constantly ventilated to reduce thermal stress on the plants, the average air temperature during the study was the same as the outdoor average. Although the daily temperature fluctuations in the greenhouse were greater than outdoors, the indoor and outdoor air temperatures were significantly correlated. The discrepancy between the soil temperature measured by our sensor and the highest measured outdoor temperatures in summer and early winter (23.04 °C and 25.03 °C, respectively) was less than 5 °C. The outdoor soil humidity showed a significant correlation with the indoor air and soil humidity (Table 5). 

### 3.6. Plant Physiological Measurements

The seasons in correspondence with the fertiliser application (T1: control—no fertiliser application, T2—fertiliser application) significantly affected the physiological parameters of the plants. With T2 treatment, the plant height, stem thickness, leaf blade, leaf area, and yield increased by 40.95%, 8.59%, 17.95%, 6.27%, and 27.27%, respectively, in winter compared to summer. Similarly, with T1 treatment, the plant height, stem thickness, leaf blade, leaf area, and yield increased by 6.88%, 12.72%, 28.26%, 2.75%, and 4.97%, respectively, in winter compared to summer in the greenhouse (Figure 6). In this study, the high fluctuations in the environmental parameters such as soil and air temperature, light intensity, and humidity in the summer season resulted in poor plant growth and yield, with or without fertiliser application. In support of these findings, previous studies have also reported that environmental parameters such as temperature and light are the most important factors that affect the quality of pumpkin rootstock seedlings’ growth process. The responses to temperature and light are an important basis for optimising the greenhouse environment. To determine the quantitative effects of temperature and light on the growth and development of pumpkin plants, regression analysis was used to establish relationships between temperature, light, and pumpkin seedling growth [29]. The results indicated that the daily average temperature showed a significant negative correlation with the development time of pumpkin rootstock seedlings, and the shoot dry weight of pumpkin rootstock seedlings increased within a certain range of the daily light integral. 

## 4. Conclusions

The integration of information and management methods, as well as a thorough understanding of the microclimatic features affecting greenhouse production and data collection, is necessary for sustainable greenhouse production. The intriguing and relevant results of our study arise from the quantitative evaluation of sensor locations based on the number of deployed sensors using time-series data. Statistical indices were used to determine the number of sensors needed to attain a given degree of accuracy, and the accuracy of the chosen sensors was also statistically assessed. The best sensor locations for identifying places with significant variations in air temperature, soil temperature, air humidity, soil humidity, and illumination were determined by focusing on the areas most affected by the external environment. Considering how many sensors were studied in summer and early winter 2022, the location of sensor A3 (the centre of the greenhouse) proved to be an ideal sensor location in summer when the complete greenhouse atmosphere was measured using a single sensor. When a single sensor was placed at A3, the MAPE of air temperature, soil temperature, air humidity, soil humidity, and illumination for summer was calculated to be 2%, 2.12%, 1.05%, 2.60%, and 0.91%, respectively. When a single sensor was placed at A3, 3.44%, 2.40%, 1.08%, 2.64%, and 3.51% were predicted for early winter, respectively. It was found that the A3 sensor accurately detected air temperature, air humidity, soil temperature, soil humidity, and illumination. To guarantee that the discrepancy between the combined trend and the reference trend is less than 1%, our research suggests installing at least three sensors at the A3 sensor position to check the deviation between the combined trend and the reference trend. 

Nevertheless, the work presents some limitations as a result of the experimental setup. The most important has to do with the approach of taking the environmental parameters data during the ripening stage of the pumpkin rather than installing the sensors before the growing stage to check the impact of the environmental parameters on seed germination, growth, and development in both seasons. Also, the present study covers only two seasons (summer and winter), which limits its significance regarding the other seasons in the study location. Future work should focus on installing the sensors in the greenhouse for plant growth and development before soil preparation to assess the impact of the environmental parameters on seed germination, growth, and development in both seasons to determine the significance of the variability imposed by the environmental parameters on the pumpkin plant. Greenhouse data collection for more prolonged periods of time will contribute to the configuration of more representative climate files for the study location. Therefore, it is recommended that the next research work in the study location should cover the four seasons to configure more representative environmental parameters and climate files and their influence on plant growth and development.

## Figures and Tables

**Figure 1 sensors-24-00143-f001:**
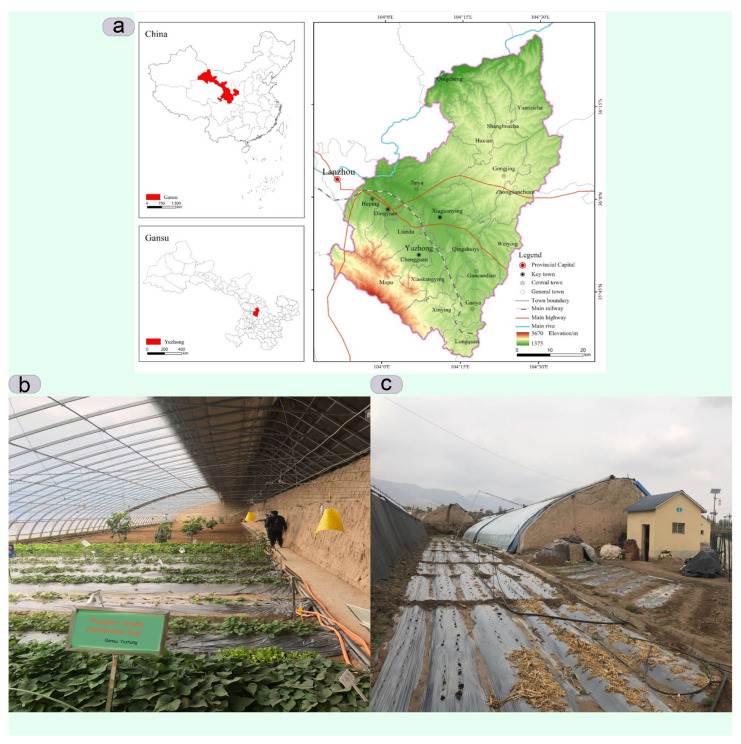
Image (**a**) shows the location of the experimental greenhouse in Yu Zhong, Lanzhou, Gansu Province, China, (**b**) shows the interior of the greenhouse used for the experiment, indicating the upper and lower vents, and (**c**) shows the exterior of the greenhouse.

**Figure 2 sensors-24-00143-f002:**
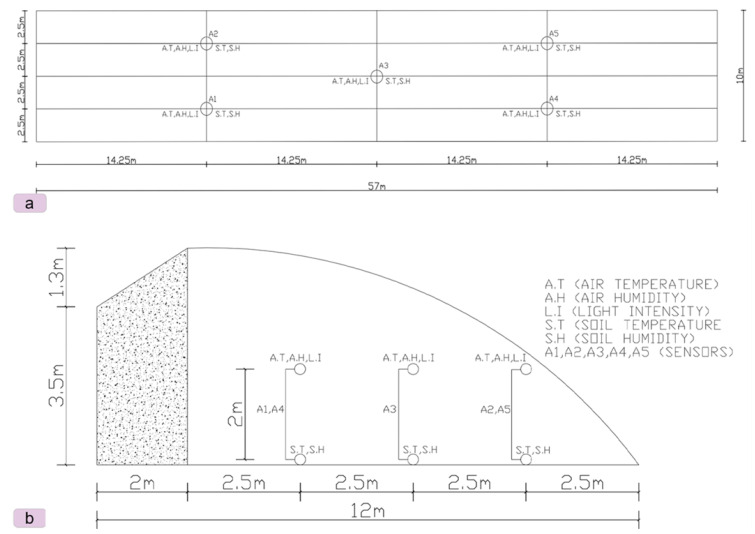
Locations of sensors for measuring the greenhouse’s internal environment. Part (**a**) shows the calculation region of the greenhouse, showing sensor placement sections, and (**b**) shows the positioning of sensors, 1 m above the ground and a side-view cross-section and 20 m from the east wall.

**Figure 3 sensors-24-00143-f003:**
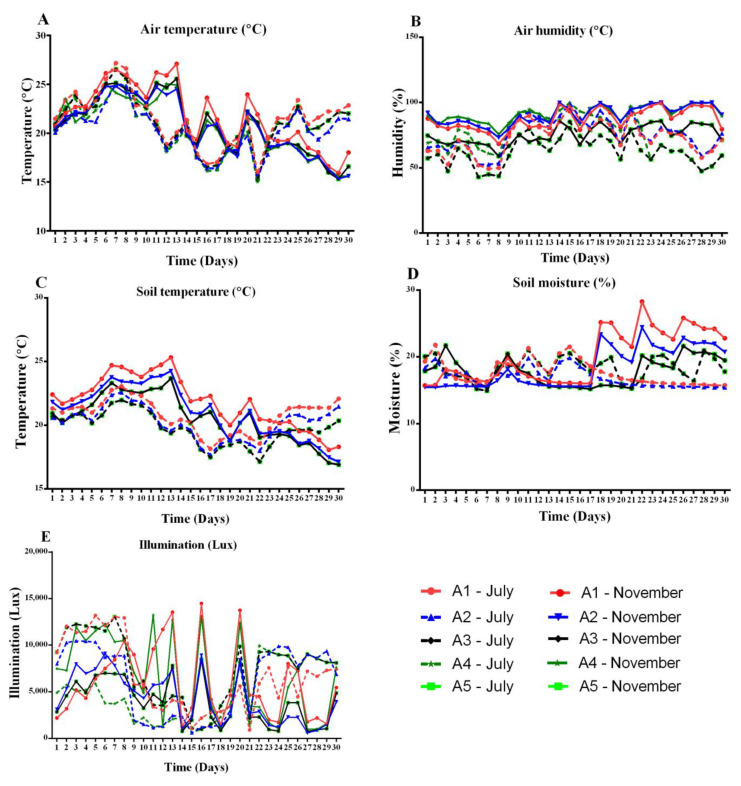
Optimal sensor simulation of (**A**) air temperature, (**B**) air humidity, (**C**) soil temperature, (**D**) soil humidity, and (**E**) illumination.

**Figure 4 sensors-24-00143-f004:**
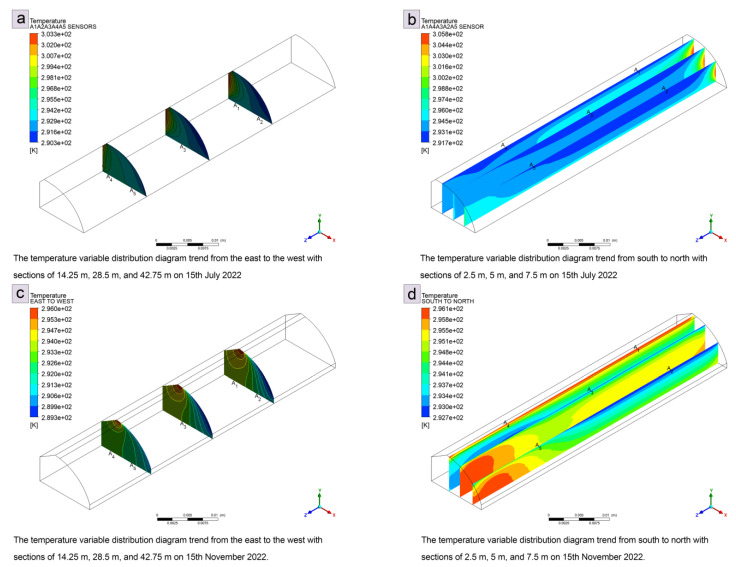
Sensors positioning simulation readings of temperature for summer and winter: (**a**) temperature variation distribution from east to west for summer, (**b**) temperature variation distribution from south to north for summer, (**c**) temperature variation distribution from east to west for early winter, and (**d**) temperature variation distribution from south to north for early winter.

**Figure 5 sensors-24-00143-f005:**
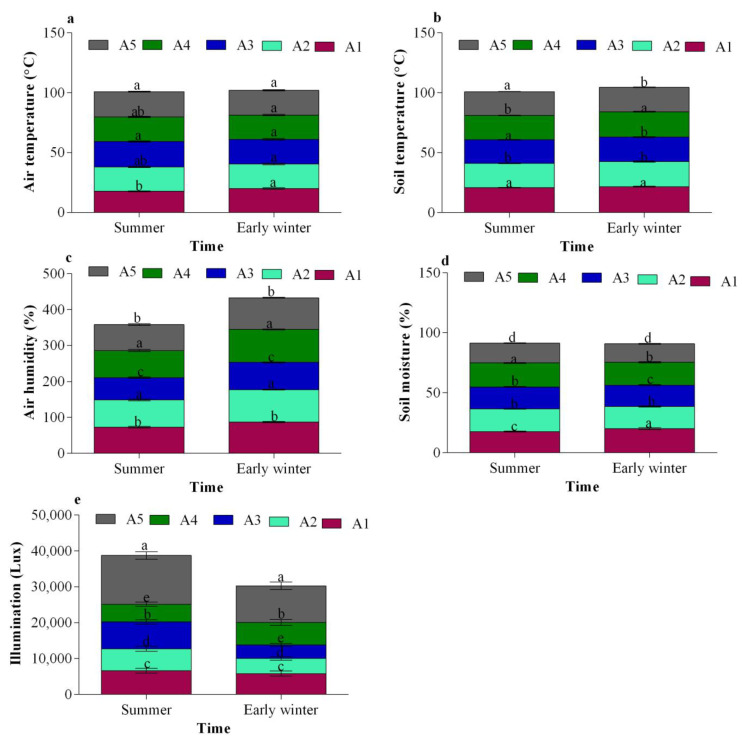
Boxplots of the sensors measuring (**a**) air temperature, (**b**) soil temperature, (**c**) air humidity, (**d**) soil humidity, and (**e**) illumination data. According to Duncan’s multiple-range tests, at *p* < 0.05, the letters above the boxplots indicate a significant difference. These various letters demonstrate that the groups are statistically distinct.

**Figure 6 sensors-24-00143-f006:**
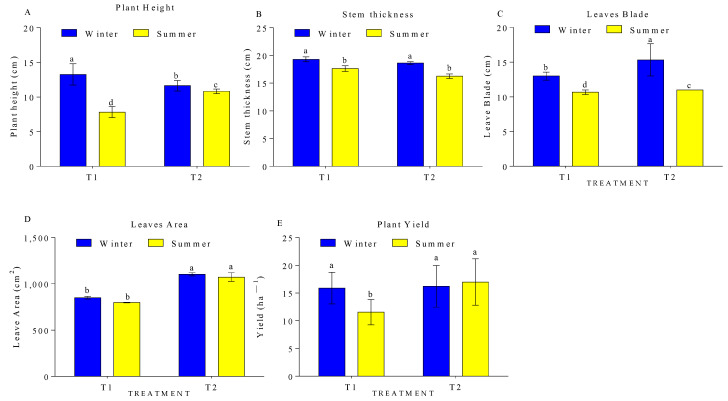
Seasonal effects on pumpkin (**A**) plant height, (**B**) stem thickness, (**C**) leaf blades, (**D**) leaf area, and (**E**) plant yield in a plastic greenhouse. Different lower-case letters indicate significant differences at *p* < 0.05 using one-way ANOVA.

**Table 1 sensors-24-00143-t001:** Descriptive statistics of inside and outside air temperature data recorded during summer and early winter 2022.

Environmental Parameter		Position of Sensors				
		A1	A2	A3	A4	A5
Air temperature (°C)	Summer average	17.71	20.35	21.10	20.72	21.10
	Standard deviation (SD)	1.98	2.29	2.63	2.47	2.63
	Standard error of mean (±)	0.36	0.41	0.47	0.44	0.47
	Winter average	20.12	20.32	20.63	20.38	20.63
	Standard deviation	3.94	3.41	3.20	3.12	3.19
	Standard error of mean (±)	0.71	0.61	0.57	0.56	0.57
Air Humidity (%)	Summer average	72.86	75.92	62.46	74.91	72.33
	Standard deviation	12.80	12.90	10.95	15.48	13.74
	Standard error of mean (±)	2.30	2.32	1.97	2.78	2.46
	Early winter average	87.10	90.79	75.19	91.67	88.40
	Standard deviation	8.26	7.58	7.08	6.38	8.06
	Standard error of mean (±)	1.48	1.36	1.27	1.14	1.45
Soil Temperature (°C)	Summer average	20.85	20.27	19.80	20.27	19.79
	Standard deviation	1.35	1.32	1.30	1.33	1.30
	Standard error of mean (±)	0.24	0.24	0.23	0.24	0.23
	Winter average	21.73	20.89	20.42	21.15	20.42
	Standard deviation	2.15	2.20	1.99	1.93	1.99
	Standard error of mean (±)	0.39	0.40	0.36	0.34	0.36
Soil humidity (%)	Summer average	17.71	18.89	18.36	19.79	16.58
	Standard deviation	1.98	1.43	1.75	1.64	1.64
	Standard error of mean (±)	0.36	0.26	0.32	0.30	0.30
	Early winter average	20.13	18.44	17.86	18.93	15.39
	Standard deviation	3.94	3.05	2.10	3.09	1.74
	Standard error of mean (±)	0.70	0.55	0.38	0.55	0.31
Illumination (Lux)	Summer average	6621.90	6166.12	7423.21	4951.15	13,605.80
	Standard deviation	3711.63	3719.95	3597.51	3083.38	5763.83
	Standard error of mean (±)	666.63	668.12	646.13	553.79	1035.21
	Early winter average	5825.96	4221.70	3798.98	6229.10	10,201.64
	Standard deviation	3938.61	2720.18	2476.91	4494.80	5838.62
	Standard error of mean (±)	707.40	488.56	444.87	807.29	1048.65

**Table 2 sensors-24-00143-t002:** The mean-square error (MSE), root-mean-square error (RMSE), and mean percentage error (MAPE) statistics are used to evaluate the precision of the soil humidity, soil temperature, air humidity, air temperature, and illumination data collected by every sensor concerning the standard pattern.

Environmental Parameter		Position of Sensors				
Air temperature (°C)	Summer	A1	A2	A3	A4	A5
	MSE	1.06	1.27	0.24	0.48	0.34
	RMSE	1.03	1.13	0.49	0.69	0.58
	MAPE (%)	3.21	4.71	1.92	2.53	2.08
	Early winter					
	MSE	3.27	2.50	0.61	1.02	0.84
	RMSE	1.81	1.58	0.78	1.01	0.92
	MAPE (%)	6.51	7.14	3.28	3.74	3.53
Soil temperature (°C)	Summer					
	MSE	0.34	0.53	0.43	1.23	0.34
	RMSE	0.58	0.73	0.66	1.11	0.58
	MAPE	2.28	2.95	2.07	4.22	2.17
	Early winter					
	MSE	1.29	0.97	0.44	1.31	0.48
	RMSE	1.13	0.98	0.67	1.14	0.69
	MAPE	4.78	3.62	2.27	4.39	2.61
Air humidity (%)	Summer					
	MSE	5.16	4.97	0.33	1.55	1.08
	RMSE	2.26	2.22	0.58	1.24	1.04
	MAPE	2.32	2.21	0.81	1.50	1.29
	Early winter					
	MSE	5.01	9.8	0.72	3.33	1.57
	RMSE	2.25	3.13	0.85	1.83	1.25
	MAPE	2.12	2.92	0.89	1.83	1.27
Soil humidity (%)	Summer					
	MSE	1.04	0.59	0.31	4.58	0.30
	RMSE	1.02	0.77	0.56	2.14	0.54
	MAPE	5.18	3.52	2.57	8.70	2.71
	Early winter					
	MSE	2.63	1.90	0.25	3.76	0.40
	RMSE	1.62	1.38	0.50	1.94	0.63
	MAPE	4.49	5.59	2.39	7.85	2.70
Illumination (Lux)	Summer					
	MSE	311,412.90	122,500	3219.43	276,781.21	3252.42
	RMSE	558.04	350	56.74	526.1	57.03
	MAPE	6.82	11.95	1.26	19.59	0.56
	Early winter					
	MSE	250,000	67,600	3995.50	202,815.12	64,526.16
	RMSE	500	260	63.21	450.35	254.02
	MAPE	14.24	10.79	3.04	16.42	3.97

**Table 3 sensors-24-00143-t003:** The error-based and entropy-based approaches were used to select the best location for sensors for the summer of 2022. Background color represents the best sensor locations and number of sensors used for Error-Based and Entropy-Based Method approaches.

Error-Based Method								Entropy-Based Method					
Air Temperature (°C)								Air Temperature (°C)					
	Sensor Location							Sensor Location					
Number of sensors		1	2	3	4	5	Number of sensors		1	2	3	4	5
	1							1					
	2							2					
	3							3					
	4							4					
	5							5					
Soil temperature (°C)								Soil temperature (°C)					
		1	2	3	4	5			1	2	3	4	5
Number of sensors	1						Number of sensors	1					
	2							2					
	3							3					
	4							4					
	5							5					
Air humidity (%)								Air humidity (%)					
		1	2	3	4	5			1	2	3	4	5
Number of sensors	1						Number of sensors	1					
	2							2					
	3							3					
	4							4					
	5							5					
Soil humidity (%)								Soil humidity (%)					
		1	2	3	4	5			1	2	3	4	5
Number of sensors	1						Number of sensors	1					
	2							2					
	3							3					
	4							4					
	5							5					
Illumination (Lux)								Illumination (Lux)					
		1	2	3	4	5			1	2	3	4	5
Number of sensors	1						Number of sensors	1					
	2							2					
	3							3					
	4							4					
	5							5					

**Table 4 sensors-24-00143-t004:** The optimal position for sensors in the early winter of 2022 is chosen using error-based and entropy-based methods. Background color represents the best sensor locations and number of sensors used for Error-Based and Entropy-Based Method approaches.

Error-Based Method								Entropy-Based Method					
Air Temperature (°C)								Air Temperature (°C)					
	Position of Sensor							Position of Sensor					
Number of sensors		1	2	3	4	5	Number of sensors		1	2	3	4	5
	1							1					
	2							2					
	3							3					
	4							4					
	5							5					
Soil temperature (°C)								Soil temperature (°C)					
		1	2	3	4	5			1	2	3	4	5
Number of sensors	1						Number of sensors	1					
	2							2					
	3							3					
	4							4					
	5							5					
Air humidity (%)								Air humidity (%)					
		1	2	3	4	5			1	2	3	4	5
Number of sensors	1						Number of sensors	1					
	2							2					
	3							3					
	4							4					
	5							5					
Soil humidity (%)								Soil humidity (%)					
		1	2	3	4	5			1	2	3	4	5
Number of sensors	1						Number of sensors	1					
	2							2					
	3							3					
	4							4					
	5							5					
Illumination (Lux)								Illumination (Lux)					
		1	2	3	4	5			1	2	3	4	5
Number of sensors	1						Number of sensors	1					
	2							2					
	3							3					
	4							4					
	5							5					

**Table 5 sensors-24-00143-t005:** Correlation between the indoor and outside environmental climates.

A3 Sensor	1	2	3	4	5	6	7	8
Outside soil humidity	1							
Inside soil humidity	−0.107	1						
Outside air humidity	−0.743	−0.129	1					
Inside air humidity	0.919 **	0.109	−0.917 *	1				
Outside air temperature	−0.847 *	0.082	0.930 **	−0.897 *	1			
Inside air temperature	−0.910 *	−0.046	0.476	−0.779	−0.887 *	1		
Outside soil temperature	0.099	−0.104	−0.071	0.188	0.148	−0.149	1	
Inside soil temperature	−0.004	−0.153	0.118	−0.077	−0.115	−0.054	−0.649	1

**. Correlation is significant at the 0.01 level (2-tailed); *. Correlation is significant at the 0.05 level (2-tailed).

## Data Availability

Data used in this study will be made available upon request.

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
