# Peer review of "The Influence of External Parameters on the Ripeness of Pumpkins"

_sensors, 2023, doi:10.3390/s24010143_

Round 1

Reviewer 1 Report

Comments and Suggestions for Authors

The paper is devoted to determining the optimal number and location of sensors in a greenhouse to assess the influence of external conditions on the growth of pumpkins. To increase agricultural productivity and automate the work of greenhouses, this work is very relevant. In addition, the use of time series data for the evaluation of sensor parametrs is currently significant. However, the submission of the paper needs to be improved before it is accepted for publication. Namely:

1. The title of the article is difficult to understand and needs to be improved: “the influence of external parameters on pumpkins” needs to be specified, for example, “on the process of growing pumpkins”, “on the ripeness of pumpkins”, etc.

2. The abstract is written inconsistently.Results, conclusions and methods are mixed. The conclusions are practically not reflected in the annotation; what A3 is in the annotation is completely unclear. It is better to replace A3 with a verbal description. Structure the abstract into sections of the article and add conclusions.

3. The goal states (lines 78-80) that the influence of external factors on the growth of pumpkins is being considered, but the methods and materials do not describe how the growth of pumpkins was studied or what indicators were used to describe it. The results also don't include information about pumpkin growth. You need to add this information in the appropriate sections.

4. In materials and methods. Please explain (line 130) the collection of material for analysis was carried out for one month (31 days) in July in the summer and November in the winter? paragraph 2.4 does not describe how the reference trend was selected and how it is related to the growth of pumpkins or other parameters reflecting optimal conditions for growing pumpkins. Maybe these works were carried out earlier, then it is appropriate to quote them here. It is also worth explaining in this section where the formulas for calculating the parameters MSE, RMSE, etc. come from.

5. Section Results contains not only results, but reasoning and discussion. Change the title of the section or move the discussion points and description of the results into a separate section.

6. In the conclusion (line 498) it is indicated that the optimal use is of three sensors, but there is no clarification on their location. Also, the conclusion lacks information about the possible impact of such monitoring on the growth of pumpkins, which is stated in the purpose of the work.

7. There are still a number of minor shortcomings that need to be corrected:

1. "Table 4" is listed twice in the title of table (line 454 and line 468), follow the order of numbers.

2. Line 457 apparently remains from the template and is not needed in the text.

3. Line 473, section name, extra numbers.

Please check the entire text for such errors.

Comments on the Quality of English Language

Correct the style and readability of the text. The sentences are difficult to understanding.

Reviewer 2 Report

Comments and Suggestions for Authors

The present Ms "Measuring the effect of environmental parameters on pumpkin using optimal sensor placement technique in greenhouse" aims to determine the most suitable sensors for greenhouses and optimize the indoor environment. Ms presents significant contribution in the field. However substantial revision is required before further consideration. 

  Therefore it needs major revision. Authors should focus on the following points:   1. The Introduction section of Ms should include the limitations of previous works that insisted authors to conduct present work.    2. The novelty of the present work is not precisely indicated in the last paragraph of Introduction section.    3. Authors can add a brief info regarding the implications of statistical methods used for comparative analysis.   4. Authors should state the limitation of this study at the end of conclusion.   5.  L30. Italicize the scientific name.    6.L46. Correct the presentation of CO2   7.L48-49 Sentence is not clear and it should be rewritten.   8. L181 Please specify the type of ANOVA used.   9. L375-377 Sentence is not clear.    10. L494-495 Please check the sentence for it's accuracy.   11. Clear outcome and further reccomendation are missing in this Ms.             Comments on the Quality of English Language Authors should recheck the entire Ms for typos and english language.   

Round 2

Reviewer 2 Report

Comments and Suggestions for Authors

The manuscript has been adequately revised.